# The Protective Effect of Mangiferin on Formaldehyde-Induced HT22 Cell Damage and Cognitive Impairment

**DOI:** 10.3390/pharmaceutics15061568

**Published:** 2023-05-23

**Authors:** Fan Chen, Na Wang, Xinyan Tian, Juan Su, Yan Qin, Rongqiao He, Xiaping He

**Affiliations:** 1School of Basic Medical Sciences, Dali University, Dali 671003, China; anlyerxian@foxmail.com (F.C.); wang08140827@163.com (N.W.); txybaoshan@163.com (X.T.); sujuan@dali.edu.cn (J.S.); lantingxun@126.com (Y.Q.); 2State Key Laboratory of Brain and Cognitive Science, Institute of Biophysics, Chinese Academy of Sciences, Beijing 100045, China; 3Key Laboratory of Mental Health, Institute of Psychology, Chinese Academy of Sciences, Beijing 100045, China

**Keywords:** Alzheimer’s disease, formaldehyde, mangiferin, Tau hyperphosphorylation, endoplasmic reticulum stress, glycogen synthase kinase-3β, calmodulin-dependent protein kinase II

## Abstract

Formaldehyde (FA) has been found to induce major Alzheimer’s disease (AD)-like features including cognitive impairment, Aβ deposition, and Tau hyperphosphorylation, suggesting that it may play a significant role in the initiation and progression of AD. Therefore, elucidating the mechanism underlying FA-induced neurotoxicity is crucial for exploring more comprehensive approaches to delay or prevent the development of AD. Mangiferin (MGF) is a natural C-glucosyl-xanthone with promising neuroprotective effects, and is considered to have potential in the treatment of AD. The present study was designed to characterize the effects and mechanisms by which MGF protects against FA-induced neurotoxicity. The results in murine hippocampal cells (HT22) revealed that co-treatment with MGF significantly decreased FA-induced cytotoxicity and inhibited Tau hyperphosphorylation in a dose-dependent manner. It was further found that these protective effects were achieved by attenuating FA-induced endoplasmic reticulum stress (ERS), as indicated by the inhibition of the ERS markers, GRP78 and CHOP, and downstream Tau-associated kinases (GSK-3β and CaMKII) expression. In addition, MGF markedly inhibited FA-induced oxidative damage, including Ca^2+^ overload, ROS generation, and mitochondrial dysfunction, all of which are associated with ERS. Further studies showed that the intragastric administration of 40 mg/kg/day MGF for 6 weeks significantly improved spatial learning ability and long-term memory in C57/BL6 mice with FA-induced cognitive impairment by reducing Tau hyperphosphorylation and the expression of GRP78, GSK-3β, and CaMKII in the brains. Taken together, these findings provide the first evidence that MGF exerts a significant neuroprotective effect against FA-induced damage and ameliorates mice cognitive impairment, the possible underlying mechanisms of which are expected to provide a novel basis for the treatment of AD and diseases caused by FA pollution.

## 1. Introduction

Alzheimer’s disease (AD), a central neurodegenerative disease that manifests as progressive memory loss, has an intricate pathogenesis and a hitherto unknown etiology. AD afflicts more than 25 million people worldwide, as well as their families, and this number is still increasing [1]. With this comes the increased cost of treating and caring for AD patients. This economic burden has prompted the ongoing exploration and development of drugs for the treatment of AD. The accumulation of extracellular β-amyloid (Aβ) and the hyperphosphorylation of intracellular Tau are classical pathological features of AD [2,3,4]. However, drugs developed that target Aβ or Tau alone are not satisfactory, suggesting that a deeper understanding of the pathogenesis of AD should be explored to reveal more comprehensive approaches to delay or prevent the development of AD.

Formaldehyde (FA) is a toxic aldehyde that can cross the blood-brain barrier (BBB) and is able to induce cognitive impairment and neurotoxicity, including the deposition of Aβ and Tau hyperphosphorylation in vitro and in vivo [5,6], and is therefore considered a new pathogenic factor in AD. Thus, it has been widely used to simulate AD disease models [7,8]. In recent years, a series of studies have begun to explore the mechanism of FA-induced AD pathogenesis. As one of the markers of oxidative stress, the appearance of FA is often accompanied by an increase in ROS and lipid peroxides [5,9,10,11]. Strong oxidative stress and cytotoxicity can further lead to DNA lesions [12], endoplasmic reticulum stress (ERS) [11], and the impairment of mitochondrial function [13]. Glucose-regulated protein 78 (GRP78, also known as Bip) and CCAAT/enhancer binding protein (C/EBP) homologous protein (CHOP) expression, biomarkers of ERS, were found to increase after exposure to FA [14,15,16]. The consequence of persistent ERS is that the unfolded protein response (UPR) overreacts, which in turn disrupts intracellular Ca^2+^ homeostasis [17] to initiate the mitochondrial apoptotic program [18]. Thus far, an increasing number of studies are beginning to elucidate the role of ERS in the pathological progression of AD [19,20]. The core of AD is a type of disease with protein misfolding and/or aggregation [21], which can cause ER dysfunction and, ultimately, ERS [22,23]. Therefore, ERS is considered to be an important link in the pathogenesis of FA-induced neurotoxicity. Some studies have also confirmed a vicious cycle between ERS and Tau hyperphosphorylation [24], but whether FA-induced Tau hyperphosphorylation is associated with ERS has not yet been thoroughly studied.

The hyperphosphorylation of Tau is widely considered a pathological hallmark of AD, with about 85 putative phosphorylation sites (p-sites) having been predicted (Ser, Thr, and Tyr) along the amino acid sequence of Tau; however, not all the p-sites are disease-associated or toxic [25,26]. The main p-sites used for characterizing pathological AD Tau by p-site specific Tau antibodies include AT8 and AT100, or pT181, pS396, etc. [27]. It has been shown that the elevated concentration of pT181 in CSF is a highly specific pathological marker for AD, which remained normal in other types of dementias [28]. Moreover, plasma pT181 can discriminate AD from several other neurodegenerative diseases with high performance, and predict cognitive decline and hippocampal atrophy over a period of 1 year [28,29]. Therefore, pT181 is an important indicator in characterizing the pathological changes in AD. The reversible nonpathological phosphorylation of Tau depends on the interaction of Tau kinases and phosphatases, and the changes in any activity could lead to elevated Tau phosphorylation. As described by our previous work, one of the pathways by which FA leads to Tau hyperphosphorylation is achieved by up-regulating the expression of glycogen synthase kinase-3β (GSK-3β) and calmodulin-dependent protein kinase II (CaMKII), both pivotal Tau kinases in AD, and down-regulating PP2A [30]. In addition, it has been shown that ERS leads to AD-related Tau hyperphosphorylation via the activation of GSK-3β [24]. The inhibition of GSK3 has been shown to effectively protect neuronal cells from ERS-induced apoptosis [31]. Furthermore, research has shown that CaMKII activation participates in regulating the ERS apoptosis pathway [32]. Therefore, elucidating the crosstalk between FA-induced ERS and Tau kinases may be a unique idea for investigating the underlying mechanism of AD and developing therapeutic drugs.

Mangiferin (MGF) is a chemically stable polyphenolic C-glucosyl-xanthone with a chemical structure comprising aglycone and glycone [33]. It has strong effects, such as its antioxidant, anti-inflammatory, anti-lipid peroxidation, and anti-apoptosis effects, and shows a wide range of therapeutic potential in resisting neurodegenerative diseases, for instance AD, via different molecular pathways [33,34,35]. A study on SAMP8 mice showed that MGF significantly restored learning and memory impairment, and reduced pathological injury in the hippocampus, by modulating lipid oxidation and Aβ deposition in the brain [35]. In an APP/PS1 animal model, the long-term treatment of MGF improved central pathology and cognitive deficits by diminishing the inflammatory processes [36]. In Wistar rats, MGF ameliorated intracerebroventricular-quinolinic acid-induced cognitive deficits, oxidative stress, and neuroinflammation [37]. Furthermore, MGF can resist glutamate-mediated oxidative damage in neurons by inhibiting ROS production and inhibiting intracellular Ca^2+^ overload, thus alleviating cognitive impairment [38]. Cholinergic transmission pathways are commonly impaired in various neurological diseases associated with cognitive impairment, including AD [39]. Studies have shown that both increased and decreased acetylcholinesterase (AChE) activity led to behavioral alterations and cognitive impairment [40,41]. Interestingly, MGF was found to be able to counteract cholinergic dysfunction and promote memory recovery by inhibiting AChE expression and stimulating cholinergic receptors [42,43]. These results suggest that MGF has a good therapeutic potential for AD. However, whether MGF protects neurons from FA-induced injury has not been elucidated. In this study, we investigated the neuroprotective effect of MGF on FA-treated C57/BL6 mice and HT22 cells, aiming to clarify the neuroprotective effect of MGF on FA-induced neurotoxicity and the underlying mechanisms to open up new therapeutic pathways for the treatment of AD.

## 2. Materials and Methods

### 2.1. Experimental Animals

Eight-week-old male C57/BL6 mice were obtained from Zhili Biotechnology Co., Ltd. (Kunming, Yunnan, China). The mice were housed at a temperature of 22–26 °C, relative humidity of 40–70%, light/dark cycle of 12 h light/12 h dark (8:00 a.m. to 8:00 p.m.), and allowed access to water and food ad libitum. All experiments were conducted in accordance with the relevant regulations of the Dali University Laboratory Animal Center for the management and handling of laboratory animals, as well as international guidelines on the ethical use of animals.

### 2.2. Experimental Cells

The mouse hippocampal neuronal cell line HT22 is widely used to construct in vitro models of neurodegenerative diseases. The HT22 cells used in this study were kindly provided by Dr. Zhiting Gong, School of Basic Medical Sciences, Dali University.

### 2.3. Reagents

FA aqueous solution (37% *w*/*v*) was from Sigma Aldrich (USA); methanol solution (99.5%) was from Xilong Scientific Co., Ltd. (Shenzhen, China). Mangiferin (MGF) standards (≥98%) were obtained from Dibai Biotechnology Co., Ltd. (Shanghai, China); 4-phenylbutyric acid (4-PBA) was from Aladdin (Shanghai, China); SB216763 and KN-93 phosphate standards were from Yuanye Bio-Technology (Shanghai, China); Dulbecco’s modified Eagle’s medium (DMEM)/F12, phosphate buffer solution, penicillin streptomycin mixture (100×), trypsin solution (0.25%), and special grade fetal bovine serum (FBS) were products from Biological Industries (Israel); dimethyl sulfoxide (DMSO, cell culture grade) was obtained from Solarbio (Beijing, China); Cell Counting Kit-8 (CCK8) came from Proteintech (Wuhan, China); BCA protein assay kit, RIPA lysate, DCFH-DA (ROS detection probe), and DAPI staining solution were from Share-bio (Shanghai, China); and calcium content chromogenic assay kit and Mito-Tracker Red CMXRos (mitochondrial red fluorescent probe) were obtained from Beyotime Biotechnology (Shanghai, China). Citric acid antigen repair solution, hematoxylin staining solution, and DAB color reagent were from Servicebio technology Co., Ltd. (Wuhan, China). Other analytical grade reagents, unless otherwise specified, were obtained from a local reagent vendor.

### 2.4. Antibodies

Anti-β-actin (SB-AB0035) and goat anti-rabbit IgG horseradish peroxidase (HRP)-conjugated secondary antibodies (SB-AB0101) were from Share-bio (Shanghai, China); CaMKII (ab52476) and anti-total GSK3β (ab93926) were obtained from Abcam (Cambridge, UK); anti-p-GSK3β (Tyr216) (R24515), anti-total Tau (R25862), and anti-p-Tau (Thr181) (310192) came from ZEN-BIO (Chengdu, China); goat anti-mouse IgG horseradish peroxidase (HRP)-conjugated secondary antibodies (SA00001-1), anti-GRP78 (66574), and anti-CHOP (66741) were obtained from Proteintech (Wuhan, China).

### 2.5. Cell Culture

HT22 cells were cultured in DMEM/F12 medium supplemented with 10% FBS and 1% penicillin streptomycin mixture. Incubation conditions: 37 °C, 5% CO_2_. The culture medium of the cells was changed every other day, and the cells were passaged when they reached 80% to 90% confluence.

### 2.6. Experimental Design

HT22 cells were divided into the following groups: control group (cells without any treatment), vehicle control group (0.4% DMSO + FA co-treatment); FA group (FA applied at an appropriate concentration according to each experimental scale); MGF group (MGF applied at an appropriate concentration according to each experimental scale); and FA + MGF group (MGF applied at an appropriate concentration according to each experimental scale). Inhibitor groups: FA + 4PBA (1 mM) group, FA + KN-93 (5 μM) group, and FA + SB216763 (20 μM) group.

The AD mouse model was constructed according to the method of Yang et al. [8] with slight modifications. According to the feature that methanol could be metabolized into FA in vivo, 8-week-old mice were intragastrically administered with 3.8% aqueous methanol for 6 weeks. The mice were randomly divided into five groups of 12 mice each based on body weight: control group (pure water); FA group (3.8% methanol aqueous); low-dose group (3.8% methanol + 5 mg/kg MGF aqueous); medium-dose group (3.8% methanol + 20 mg/kg MGF aqueous); and high-dose group (3.8% methanol + 40 mg/kg MGF aqueous). Methanol and aqueous MGF were intragastrically administered once a day for 6 weeks. Then, the Y-maze test and novel object recognition test were conducted to evaluate spatial learning ability and long-term memory after 1 week of recovery.

### 2.7. Molecular Docking

The mol2 file of MGF was obtained from the TCMSP database (Lab of Systems Pharmacology, Xi’an, China). PDB files for GRP78 and CHOP structures were obtained from the RCSB Protein Data Bank [44] by setting the organism as “Homo sapiens”. The 2D structures of MGF and associated protein 3D structures were imported into AutoDock Tools (version 1.5.7) [45] for molecular docking, and docking results were visualized with PyMOL software. The binding energy and root mean square deviation (RMSD) were used to assess the reliability of the docking results. A binding energy <−1.2 kcal/mol [45] and RMSD <2 Å [46] indicate that the compound can spontaneously bind to the target protein, and the lower the binding energy is, the better the binding affinity of the ligand to the relevant target protein.

### 2.8. Cytotoxicity Assay

Cell Counting Kit-8 (CCK-8) was used to determine cell viability according to the manufacturer’s instructions. Briefly, HT22 cells were seeded in a 96-well plate (Corning, CA, USA) at a density of 1.5 × 104 cells/well and cultured for 24 h to ensure cell attachment. The medium was then replaced with DMEM/F12 (without FBS) containing FA (0.1 to 0.9 mM), MGF (25 to 250 µM), and DMSO (0.1 to 0.5%), respectively. The optical absorbance at 450 nm was measured on a spectrum plate reader (Bio-Red, Hercules, CA, USA) and cell viability was expressed as a percentage of viable cells relative to the untreated control group.

Changes in the morphology of cells were observed with the following method: HT22 cells (2.0 × 10^5^ cells/well) were seeded in a 12-well plate using cell slides and cultured for 24 h to ensure cell attachment. Then, the cells were co-treated with MGF (25 to 250 µM) and FA (0.5 mM) for 4 h. The cells were fixed with 4% paraformaldehyde (PFA) for 10 min and then observed using an upright fluorescence microscopy imaging system (BXS3, OLYMPUS, Tokyo, Japan).

### 2.9. Ca^2+^ Concentration Determination

Intracellular Ca^2+^ concentration was detected by calcium colorimetry kit (Beyotime, Haimen, China). Briefly, according to the manufacturer’s instructions, HT22 cells were seeded in a 6-well plate at a density of 4 × 10^5^ cells/well and, after a 24 h attachment period, the cells were co-treated with DMSO (0.4%), MGF (25 to 250 µM), and FA (0.5 mM) for 4 h, respectively. The cells were fully lysed using the lysate supplied with the kit, the supernatant was centrifuged and collected, and the display reagent was then added to the supernatant and incubated in the dark for 15 min. The supernatant was centrifuged, and the absorbance was detected at 575 nm on a microplate reader (HEALES Co., Ltd., Shenzhen, China). The final Ca^2+^ concentration was expressed as a percentage relative to the control group concentration.

### 2.10. Determination of Oxidative Damage

For detection of mitochondrial damage according to the manufacturer’s instructions, HT22 cells (2.0 × 10^5^ cells/well) were seeded in a 12-well plate using cell slides and cultured for 24 h to ensure cell attachment. After co-treatment with MGF (250 µM) and FA (0.5 mM) for 4 h, the medium was discarded and the cells were incubated with 100 nM Mito Tracker Red CMX Ros (mitochondrial red fluorescent probe, dissolved in FBS-free medium) at 37 °C in the dark for 30 min. After gentle rinsing with warm FBS-free medium, the cells were fixed with 4% PFA for 10 min and permeabilized with 0.2% Triton X-100 for subsequent DAPI staining. Red fluorescence intensity was observed in a fluorescence microscopy imaging system (BXS3, OLYMPUS, Tokyo, Japan).

To detect intracellular ROS content, HT22 cells (2.0 × 10^5^ cells/well) were seeded in a 12-well plate using cell slides and cultured for 24 h to ensure cell attachment. After co-treatment with MGF (250 µM) and FA (0.5 mM) for 4 h, the medium was discarded and the cells were incubated with 10 µM DCFH-DA (ROS green fluorescence detection probe, dissolved in FBS-medium) at 37 °C in the dark for 30 min. After gentle rinsing with warm phosphate buffer containing calcium and magnesium ions, the green fluorescence intensity was observed in a fluorescence microscopy imaging system (BXS3, OLYMPUS, Tokyo, Japan).

### 2.11. Western Blotting Assays

For Western blotting assays, cells were seeded at 4 × 10^5^ cells/well in a 6-well plate and were co-treated with MGF (25 to 250 µM) and FA (0.5 mM) for 4 h. Western blotting procedures were performed as described in the literature [47], with small modifications. Briefly, whole-cell lysates were prepared by incubation of cells in RIPA buffer supplemented with protease and phosphatase inhibitor cocktail according to the manufacturer’s instructions. The protein content of the samples was determined using a BCA protein assay kit. Then, 20 µg of protein samples were mixed with 5× sodium dodecyl sulfate-polyacrylamide gel electrophoresis (SDS-PAGE) buffer and heated at 98 °C for 10 min, separated by SDS-PAGE gel, and then transferred to polyvinylidene fluoride membranes (Millipore, Burlington, MA, USA). After blocking with bovine serum albumin (5% *w*/*v*) for 1 h, the membranes were incubated with primary antibodies (β-actin, 1:6000; Tau, 1:2000; p-Tau (Thr181), 1:1500; GSK-3β, 1:2000; p-GSK-3β (Tyr216), 1:1500; CaMKII, 1:2000; GRP78, 1:5000; CHOP, 1:5000) for 1.5 h at 37 °C or overnight at 4 °C. The membranes were slowly washed 3 times for 10 min each in Tris-HCl buffered saline (with 1% Tween-20) (TBST). Subsequently, the membranes were incubated with the corresponding anti-rabbit/mouse IgG HRP-conjugated secondary antibody (1:10,000) for 1 h at room temperature. The membranes were again washed 3 times for 10 min in TBST. The optical density was measured by an automatic chemiluminescence image analyzer (Shanghai Bio-Tech Co., Ltd., Shanghai, China). ImageJ software package (National Institute of Health, Besthada, Rockville, MD, USA) was used to quantify the optical density. The individual band intensities were normalized to the intensity of β-actin bands.

### 2.12. Y-maze Test

The Y-maze was used to assess the spatial memory ability of mice [48,49]. At the beginning of the experiment, animals were transferred to the Y-maze apparatus. The device is divided into three different arms (start arm, novel arm, and other arm). The start and other arms were turned on and the novel arm was turned off during the training phase, allowing the mice to explore for 10 min. Then, 2 h after the end of training, all three arms were left open and the mice were allowed to explore freely for 5 min, and the time spent exploring each arm separately was recorded. Immediately after each round of testing, the mice were removed and returned to their original cages, and the urine and excreta of the mice in the maze were cleaned and then sprayed with 75% alcohol to eliminate possible interference from the residual odor of the mice. The analysis of data outcomes was based on the sum of time spent exploring the novel arm during the 5 min of the test phase.

### 2.13. Novel Object Recognition Test

Novel object recognition (NOR) experiments were used to assess cognitive abilities in long-term memory in mice [50,51]. At the beginning of the experiment, the animals were transferred to the NOR apparatus. The device consisted of an open field of a cube (40 cm × 40 cm × 40 cm) in which two identical objects were placed symmetrically in the diagonal. The mice were placed in the open field to move freely for 10 min, and the exploration time of the mice for the two objects was recorded. Then, 24 h after the end of the training, a familiar object in the open field was replaced by a novel object, and the mice were allowed to explore it freely for 5 min. The exploration time of the familiar object and the novel object was recorded. After each round of testing, the device was cleaned and disinfected with 75% alcohol, as in Y-maze test. Experimental data were recorded using Visutrack software. Preference performance in the NOR was assessed using the preference index (PI) during the training phase, and memory performance in the NOR was assessed using the discrimination index (DI) during the testing period. PI is the percentage of time spent exploring the novel object over the total time spent exploring two objects; DI is the difference in time spent exploring novel and familiar objects during testing as a percentage of the total time spent exploring both objects.

### 2.14. Immunohistochemistry

The mice brain tissues were fixed with 4% PFA at room temperature (RT), dehydrated with 50% sucrose solution, embedded in paraffin, and cut into 4 μm slices. The sections were deparaffinized and washed with water. Antigen repair was performed using citric acid antigen repair solution at PH = 6 (boiling for 10 min), and after cooling the sections were washed in PBS (3 times /5 min). The sections were incubated in 3% hydrogen peroxide at RT in the dark for 25 min, after which the sections were washed (3 times/5 min) in PBS. Sections were blocked using 3% BSA for 30 min at RT. Primary antibodies (Tau, 1:500; p-Tau (Thr181), 1:100; GSK-3β, 1:200; CaMKII, 1:500; GRP78, 1:1000) were incubated overnight at 4 °C, and sections were washed in PBS (3 times/5 min) before incubation with secondary antibodies at RT (1:200, 50 min). The sections were washed in PBS (3 times/5 min) and then stained with 3,3′-diaminobenzidine (DAB). Nuclei were counterstained using hematoxylin for about 3 min. Immunohistochemical staining results were observed under a light microscope (BXS3, OLYMPUS, Tokyo, Japan) after the sections were dehydrated and sealed. ImageJ software was used to analyze the positive rate of staining.

### 2.15. Statistical Analysis

The experiments in this study were independently repeated no less than three times, and data were analyzed using GraphPad Prism (Version 8.0.2, San Diego, CA, USA) software and expressed as the mean ± standard error of the mean (SEM). All the data were normally distributed (Kolmogorov–Smirnov tests: all *p* > 0.05) and were analyzed by one-way ANOVA. Post hoc comparisons were performed using Student–Newman–Keuls. *p* < 0.05 was considered statistically significant.

## 3. Results

### 3.1. MGF Protects HT22 Cells from FA-Induced Neurotoxicity

To study the mechanisms underlying the protective effect of MGF against FA-induced neurotoxicity, HT22 cells were employed. FA induced cell death in a dose-dependent manner with an LD_50_ of approximately 0.5 mM (Figure 1a). MGF alone (25 to 250 µM) did not cause any loss of HT22 cell viability after 4 h of treatment (Figure 1b). Similarly, the viability of HT22 cells was not influenced after treatment with various concentrations of DMSO (0.1 to 0.5%), the vehicle used for MGF (Figure 1c). To determine whether MGF protected HT22 cells from FA-induced neurotoxicity, cell viability assays were performed after a co-treatment of cells with MGF (25 to 250 µM) and FA for 4 h. As shown in Figure 1d, a co-treatment of MGF dose-dependently protected cells from FA-induced cytotoxicity, with a significant effect observed at 100 µM (*p* = 0.0049). Afterwards, the morphology of HT22 cells was observed at different magnifications using a microscope, and it was found that, compared with controls, FA-treated cells had a lower cell density, and cells shrunk and exhibited fewer cellular processes and irregular borders. Compared with the FA-treated group, the MGF co-treated group had an increased cell density and cell morphology that was closer to the normal state of HT22 cells. Furthermore, cell processes were also partially reverted in the MGF co-treated group (Figure 2). To observe this reverse effect, we prolonged the incubation for 24 h, and observed that cell density and cell morphology in the MGF co-treated group returned to almost the same as that of the control group.

### 3.2. MGF Decreases FA-Induced Tau Hyperphosphorylation by Inhibiting GSK3β and CaMKII

The Western blotting results showed that Tau phosphorylation at Thr181 was significantly increased in HT22 cells exposed to 0.5 mM FA for 4 h (*p* = 0.005, Figure 3a). Co-treatment with different concentrations of MGF decreased FA-induced Tau phosphorylation at Thr181 in a dose-dependent manner (*p* = 0.028, at 100 µM of MGF). Treatment with 250 µM of MGF almost completely restored the level of FA-induced Tau phosphorylation at Thr181 to normal levels (*p* = 0.0003). However, MGF co-treatment did not significantly affect the expression of total Tau (*p* = 0.9248). This result suggests that MGF is able to inhibit FA-induced Tau phosphorylation, but it does not affect the expression of Tau itself (Figure 3b,c).

To further investigate the mechanism by which MGF decreased Tau hyperphosphorylation, the expression of GSK3β and CaMKII, kinases associated with Tau phosphorylation, was examined. As shown in Figure 3d, both the levels of phosphorylation of GSK-3β at tyrosine 216 (activation site) (*p* < 0.0001) and total GSK-3β (*p* < 0.0001) were significantly increased after 0.5 mM FA exposure for 4 h, while co-treatment with MGF significantly decreased the level of phosphorylation of GSK-3β at Tyr216 (*p* < 0.0001, at 100 µM MGF, Figure 3f) and total GSK-3β (*p* < 0.0001, at 100 µM of MGF, Figure 3e) in a dose-dependent manner. Similar to GSK-3β, FA significantly induced an elevated expression of CaMKII (*p* < 0.0001), while MGF co-treatment dose-dependently reduced CaMKII expression (*p* = 0.0133, at 100 µM of MGF, Figure 3g,h). These results illustrate that MGF reduces FA-induced Tau hyperphosphorylation by inhibiting the expression of GSK3β and CaMKII.

### 3.3. MGF Ameliorates FA-Induced ERS in HT22 cells

To further elucidate the FA-induced upstream events of GSK-3β and CaMKII upregulation, we first obtained the 2D structure of MGF from the PubChem database (Figure 4a), and MGF was used as a small molecule ligand to dock with GRP78 and CHOP to predict interactions and affinities between MGF- and ERS-related markers, respectively. The docking results showed that the molecular binding energy of MGF docked with GRP78 was −4.43 kcal/mol, and the RMSD was 1.853 Å (Figure 4b); the intermolecular binding energy of MGF docked with CHOP was −2.79 kcal/mol, and the RMSD was 0.173 Å (Figure 4c). This result suggests that MGF may exert neuroprotective effects through ERS and downstream signaling pathways.

To confirm this result, the effects of MGF on the regulation of GRP78 and CHOP activities were further investigated. The expression levels of ERS-associated proteins GRP78 (*p* = 0.0053, Figure 5a,b) and CHOP (*p* < 0.0001, Figure 5a,c) were significantly increased after exposure to 0.5 mM FA for 4 h, while co-treatment with MGF significantly decreased the expression of GRP78 (*p* = 0.0013, at 200 µM of MGF) and CHOP (*p* = 0.0001, at 200 µM of MGF) in a dose-dependent manner.

As a supplement to this result, the expression levels of GRP78 and CHOP were further measured after treatment with ERS inhibitor 4PBA. The results showed that the 4PBA treatment indeed decreased the overexpression of GRP78 (*p* = 0.0005, Figure 5d,e) and CHOP (*p* = 0.004, Figure 5d,f), which was induced by FA. Interestingly, the 250 μM MGF treatment almost achieved the effect of the 4PBA treatment (GRP78, *p* = 0.001; CHOP, *p* = 0.0073). Taken together, these data suggest that MGF may exert neuroprotective effects by suppressing ERS.

To further investigate the role of MGF in reducing FA-induced neurotoxicity, the protein levels of GSK3β, CaMKII, and Tau were measured after being treated with the ERS inhibitor 4PBA, the GSK3β inhibitor SB216763, and the CaMKII inhibitor KN93, respectively. It was found that the phosphorylation level of GSK-3β at the Tyr216 site was significantly inhibited after 4PBA treatment (*p* = 0.015, Figure 6a,b), as was the total GSK3β (*p* = 0.004, Figure 6a,c) compared with the FA group. The expression of CaMKII was also significantly suppressed by 4PBA (*p* < 0.0001, Figure 6a,d). With respect to the Tau, the 4PBA treatment significantly reduced the phosphorylation level of Tau at the Thr181 site (*p* = 0.0311, Figure 6a,e). However, the level of total Tau did not significantly change. Following treatment with the CaMKII inhibitor KN93, the expression level of CaMKII was significantly inhibited (*p* = 0.0025, Figure 6g,h), and the phosphorylation level of Tau at Thr181 was decreased significantly (*p* = 0.0027, Figure 6g,i) compared with that in the FA group. Following the addition of the GSK3β inhibitor SB216763, the expression levels of total GSK3β (*p* = 0.0005, Figure 6k,m) and the phosphorylation of GSK-3β at Tyr216 (*p* = 0.0016, Figure 6k,l) were significantly decreased, compared with that in the FA group. Interestingly, in addition to the decreased expression of *p*-Tau (*p* = 0.0017, Figure 6k,n), CaMKII levels also appeared to be partially suppressed (*p* = 0.0382, Figure 6j,p) after the SB216763 treatment. In summary, these results indicate that MGF exerts its neuroprotective effect against FA-induced neurotoxicity by suppressing crosstalk between ERS and GSK3β/CaMKII.

### 3.4. MGF Ameliorates FA-Induced Oxidative Stress and Ca^2+^ Overload in HT22 Cells

ERS and oxidative damage are closely linked events. To understand whether MGF could counteract FA-induced oxidative stress, mitochondrial function and ROS were measured in cells treated with MGF (250 µM) simultaneously with FA for 4 h in this study. The mitochondrial fluorescence intensity of cells in the FA group was significantly lower than those in the control group (*p* = 0.0003, Figure 7a,c). Co-treatment with 250 µM MGF significantly increased intracellular mitochondrial fluorescence intensity (*p* = 0.0057), indicating a reduced degree of mitochondrial damage. An analogous result was found with respect to ROS production. Cells in the FA group showed a significant increase in ROS fluorescence intensity compared with the control group (*p* = 0.0002), whereas intracellular ROS fluorescence intensity was significantly decreased in the MGF co-treated group (*p* = 0.0052) (Figure 7b,d). These results suggest that the extent of oxidative stress induced by FA in HT22 cells was partially reduced by the MGF treatment.

Moreover, an imbalance in Ca^2+^ homeostasis occurs when ER and mitochondrial functions are compromised, and the Ca^2+^ influx further exacerbates ERS. MGF at different concentrations (25 to 250 μM) was used to reveal the protective effect against FA-induced Ca^2+^ overload. The results showed that the concentration of intracellular Ca^2+^ in HT22 cells was significantly increased after 0.5 mM FA exposure (*p* < 0.0001), while MGF co-treatment dose-dependently reduced FA-induced Ca^2+^ overload (*p* = 0.002, at 25 µM of MGF) (Figure 8).

### 3.5. MGF Improves FA-Induced Spatial Memory and Cognitive Impairment in Mice

The Y-maze and novel object recognition tests were performed to investigate the neuroprotective effect of MGF against FA-induced neurotoxicity. Figure 9a shows the effects of different concentrations (5, 20, and 40 mg/kg) of MGF on spatial memory performance. Compared with the control group, the FA group had a significant reduction in the time spent exploring the novel arm (*p* < 0.0001). MGF dose-dependently increased the time spent exploring the novel arm (*p* = 0.0428, at 20 mg/kg of MGF; *p* = 0.0011, at 40 mg/kg of MGF), indicating that FA-induced spatial memory loss was improved. The effect of MGF on cognitive performance in long-term memory was also assessed using the NOR test (Figure 9b,c). In the NOR test, there was no significant preference difference between the mice in each group (Figure 9b, *p* = 0.9559). Consistent with the results obtained in the Y-maze test of spatial memory, FA exposure significantly caused cognitive impairment in long-term memory compared with control mice (Figure 9c, *p* = 0.0008); MGF dose-dependently improved cognitive dysfunction in mice (Figure 9c, *p* = 0.008, at 40 mg/kg of MGF). In summary, these results indicate that MGF improved spatial learning ability and long-term memory in mice with FA-induced cognitive impairment.

### 3.6. MGF Reduced FA-Induced Tau Hyperphosphorylation and the Expression of GRP78, GSK-3β, and CaMKII in the Brain of Mice

The results in HT22 cells were further validated in FA-induced mice. Consistent with the results obtained in HT22 cells, immunohistochemical staining results showed that, compared with the control group, FA significantly induced Tau hyperphosphorylation (*p* < 0.0001, Figure 10a,c) and increased the expression of GRP78 (*p* = 0.0005, Figure 10a,d), GSK-3β (*p* < 0.0001, Figure 10a,e), and CaMKII (*p* = 0.0053, Figure 10a,f). While the co-treatment of MGF significantly reduced the FA-induced phosphorylation of Tau (*p* = 0.032, at 20 mg/kg of MGF; *p* = 0.0008, at 40 mg/kg of MGF, Figure 10a,c), it did not significantly affect the expression of total Tau (*p* = 0.999, Figure 10a,b). Similar to p-Tau, the MGF co-treatment significantly reduced elevated expressions of GRP78 (*p* = 0.032, at 5 mg/kg of MGF; *p* = 0.0107, at 20 mg/kg of MGF; *p* = 0.0008, at 40 mg/kg of MGF, Figure 10a,d), GSK-3β (*p* = 0.0295, at 20 mg/kg of MGF; *p* = 0.0004, at 40 mg/kg of MGF, Figure 10a,e), and CaMKII (*p* = 0.0287, at 40 mg/kg of MGF, Figure 10a,f). These results taken together support our finding that MGF attenuates FA-induced Tau hyperphosphorylation and the expression of GRP78, GSK-3β, and CaMKII in the brains of mice, which results in the amelioration of cognitive impairment.

## 4. Discussion

In recent years, a growing number of studies have found that age-related in vivo FA accumulation resulting from both extrinsic and intrinsic pathways may be a risk factor in the occurrence and development of AD [5,52,53,54,55,56,57], and targeting its inhibition has been implicated as a potential AD prophylactic. In fact, an increasing number of studies are beginning to look for natural alternatives for the treatment of AD by protecting neurons against FA-induced damage [30,58,59]. The current investigation sought to find a multitargeted, low-toxicity, and naturally available drug against FA-induced neurotoxicity to open a new venue for the treatment of AD. Here, we observed that FA exposure significantly induced AD markers, including cognitive impairment, Tau hyperphosphorylation, and cell toxicity. Co-treatment with polyphenol antioxidant MGF significantly improved spatial learning ability and long-term memory in C57/BL6 mice with FA-induced cognitive impairment. It was further revealed that MGF significantly decreased FA-induced cytotoxicity and inhibited Tau hyperphosphorylation at Thr181 in a dose-dependent manner in HT22 cells. These protective effects were achieved by attenuating FA-induced ERS, as indicated by the inhibition of ERS markers GRP78 and CHOP and downstream Tau-associated kinases (GSK-3β and CaMKII) expression. In addition, MGF could markedly inhibit FA-induced oxidative damage, including Ca^2+^ overload, ROS generation, and mitochondrial dysfunction, all of which are associated with ERS.

As massive neuronal atrophy and death are very common pathological features of AD [60,61], the present study comprehensively analyzed the effects of MGF on cell viability and cell morphology. Our results showed that a co-treatment of MGF dose-dependently protected cells from FA-induced cytotoxicity, with a significant effect observed at 100 µM or greater. Furthermore, compared with the FA-treated group, cells treated with MGF had a relatively increased cell density, and the cell morphology was closer to the normal state. As our peers emphasized, GSK-3β is a pivotal kinase in AD [60,62], and dysregulated CaMKII is a key contributor to memory impairment in neurodegeneration [63]. Additionally, as described by our previous work, FA exposure significantly induced Tau hyperphosphorylation, partially by up-regulating kinases GSK-3β and CaMKII [30]; it was then hypothesized that the protective effect of MGF was exerted by reducing phosphorylated Tau by regulating the above kinases. Interestingly, it was found that MGF reduced FA-induced Tau phosphorylation at Thr181 in a dose-dependent manner, with a significant effect at 100 μM, and higher doses of MGF (250 μM) almost completely reversed phosphorylated Tau levels at Thr181, relative to the normal group. In accordance with this, the MGF treatment markedly inhibited the FA-induced expression of both GSK-3β and CaMKII. These results indicated that the regulation of these two kinases is important in the attenuation of the FA-induced hyperphosphorylation of tau protein by MGF. However, the upstream pathways for the FA-induced up-regulation of GSK-3β and CaMKII remain to be further elucidated.

Since FA has been implicated in AD pathology as a protein cross-linker known to aggregate Aβ and Tau, while ERS was originally developed in order to clear misfolded proteins to maintain ER homeostasis [64,65,66], ERS is therefore considered being an important link in FA-induced neurotoxicity. Furthermore, the aggregation of pathological Tau is considered the key to the occurrence of AD [67], and the accumulation of p-Tau often induces irreversible ERS [68], which can promote further Tau hyperphosphorylation, thereby exacerbating AD pathogenesis. Several studies have revealed a vicious cycle between p-Tau and ERS. Ho and collaborators found significant increases in ERS-related proteins, including the phosphorylation of kinase-like ER kinase (p-PERK) and the phosphorylation of eukaryotic initiation factor-2α (p-eIF2α) in the hippocampus of aged Tau transgenic mice, and they observed that ERS could induce the hyperphosphorylation of Tau at Ser396, Ser262, and Thr231 [69]. It was then hypothesized that FA-induced Tau phosphorylation, at least in part, occurs via the activation of ERS, which was confirmed by the addition of the ERS inhibitor 4PBA in the current study. That is, the 4PBA treatment significantly reduced the phosphorylation level of Tau after FA exposure, while the level of total Tau was not found to change significantly. Then, to further investigate the protective effects of MGF on FA-induced neurotoxicity, molecular docking techniques were used to predict the possibility of interaction between MGF and the ERS-related proteins, GRP78 and CHOP. The results showed a good affinity between MGF and GRP78 and CHOP. Western blotting analyses further demonstrated that MGF inhibits the elevated expression of GRP78 and CHOP induced by FA in a dose-dependent manner. Interestingly, the inhibitory effect of 250 µM GF treatment on GRP78 and CHOP almost achieved the effect of the ERS inhibitor 4PBA treatment. Taken together, these data provide the possibility for MGF to exert a neuroprotective effect by resisting ERS.

Furthermore, GSK-3β has been reported to play essential roles in the ERS signaling pathway [24,70,71]. Gomez and colleagues detected the presence of GSK-3β at the ER and mitochondria-associated membranes. They found that GSK-3β is regulated by GRP78 and can reversely regulate the inositol triphosphate receptor-mediated Ca^2+^ release pathway in ERS [70,72]. CaMKII, as a kinase regulating Tau phosphorylation, induces Tau dissociation from microtubules in a Ca^2+^-dependent manner and contributes to the formation of NFTs [63,73]. A fraction of activated CaMKII assists in enhancing the subsequent GSK-3β-catalyzed phosphorylation of Tau [74]. Additionally, the activation of CaMKII has been found to participate in regulating the ERS apoptosis pathway [32]. In the present study, ERS inhibitor 4PBA simultaneously inhibited the FA-induced upregulation of GSK-3β and CaMKII expression. The CaMKII inhibitor KN93 and the GSK-3β inhibitor SB216763 reduced the levels of Tau phosphorylation to different extents. These results together reveal that one of the pathways of FA-induced neurotoxicity and Tau hyperphosphorylation is achieved by up-regulating GSK-3β and CaMKII via the activation of ERS. Previous studies have suggested that the activation of CaMKII can cause enhanced levels of GSK-3β phosphorylation [75], and here we found that GSK-3β also affected the activation of a fraction of CaMKII. SB216763 was found to partially reduce the expression of CaMKII but it was not fully restored to normal levels. This may be due to the fact that reduced GSK-3β activity slows the release of Ca^2+^ but fails to ameliorate the autophosphorylation level of sustained CaMKII activation [63].

Additionally, studies have shown that ERS and oxidative damage interact with each other, forming a vicious circle [76]. Mitochondria and ER are directly or indirectly connected to each other via mitochondria-associated ER membranes, which jointly regulate oxidative stress. Strong, persistent ERS can instead lead to the decompensation of ER function, initiating a series of associated procedures, including Ca^2+^ efflux and cell apoptosis. When intracellular Ca^2+^ homeostasis is out of balance, excessive Ca^2+^ flows into the mitochondria, resulting in changes in mitochondrial permeability and ROS production [77]. Our results indicated that FA exposure can significantly induce oxidative stress and ROS production, which was consistent with previous studies showing that FA exerts strong cytotoxicity by inducing oxidative stress apoptosis [78,79,80]. Based on this, it was further found that FA exposure leads to impaired mitochondrial function, decreased membrane potential, and Ca^2+^ overload. Consistent with the suppressive effect of MGF on ERS, co-treatment with MGF significantly inhibited FA-induced Ca^2+^ overload, ROS generation, and mitochondrial dysfunction.

Notably, the addition of the inhibitors of ERS, CaMKII, or GSK3β alone was unable to restore phosphorylated Tau to normal levels, whereas the highest concentration of MGF (250 µM) used in this study almost completely inhibited the level of Tau phosphorylation. This further suggests that the protective effect of MGF on HT22 cells against FA-induced neurotoxicity is achieved by modulating multiple targets and pathways.

Although HT22 cells have been widely used to study neurotoxicity, cell models do not adequately simulate the cognitive impairment and behavioral and pathological changes observed with respect to AD. Thus, in this study, we used C57/BL6 mice to further comprehensively explore the protective effects of MGF on FA-induced neurotoxicity. It has been reported that MGF had the ability to penetrate the BBB [81], and oral administration of MGF for 7 days reduced neuronal death in the hippocampal CA1 region of gerbils after ischemia-reperfusion injury [82]. In the brains of SMAP8 mice, MGF could significantly reduce the level of Aβ, repair functionally damaged mitochondria, and promote the recovery of the cerebral cortex and hippocampus [83]. In addition, MGF has been shown to significantly improve memory and cognitive impairment [61,84] and inhibit ERS-related oxidative stress by reducing IRE1α phosphorylation and ROS production [85]. The results of the present study found clear support for the neuroprotective effect of MGF on memory and cognitive impairment, which showed significant improvements in spatial learning and memory performance in FA-induced C57/BL6 AD mice.

Consistent with the results obtained in HT22 cells, FA exposure is able to significantly induce Tau hyperphosphorylation and increase the expression of GRP78, GSK-3β, and CaMKII in the brains of mice, while the co-treatment of MGF significantly reduced the phosphorylation of Tau and the expression of GRP78, GSK-3β, and CaMKII were up-regulated by FA. These results taken together suggest that the amelioration of cognitive impairment was achieved by reducing FA-induced Tau hyperphosphorylation, and the expression of GRP78, GSK-3β, and CaMKII in the brains of mice.

In summary, the present study provides the first evidence that MGF exerts a significant neuroprotective effect against the FA-induced cognitive impairment of mice and HT22 cell damage by repressing the crosstalk between ERS and downstream Tau-associated kinases (GSK-3β and CaMKII), which is considered a potential therapeutic target in the treatment of AD and diseases caused by FA pollution. FA has been recognized as one of the most serious environmental pollutants which can cause multi-system damage, including the respiratory, cardiovascular, immune, reproductive, and nervous systems. The results on the protective effect of MGF against FA-induced neurotoxicity shed light on the prevention and treatment of injuries caused by short- or long-term exposure to FA. Therefore, whether MGF can counteract FA-induced damage to other systems besides nervous system deserves further investigation in the future.

## Figures and Tables

**Figure 1 pharmaceutics-15-01568-f001:**
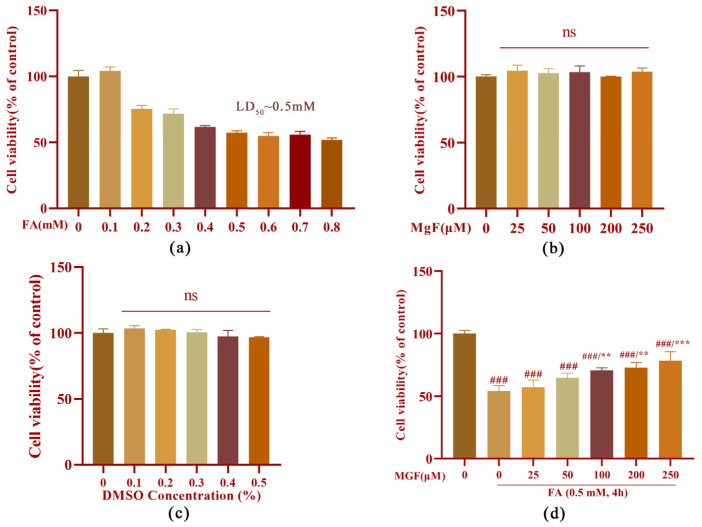
Protective effect of MGF against FA-induced cytotoxicity in HT22 cells. (**a**) Cell viability of HT22 cells treated with FA (0.1 to 0.9 mM) alone for 4 h; (**b**) cell viability of HT22 cells treated with MGF (25 to 250 µM) alone for 4 h; (**c**) cell viability of HT22 cells treated with DMSO (0.1 to 0.5%) alone for 4 h; (**d**) cell viability of HT22 cells co-treated with FA (0.5 mM) and MGF (25 to 250 µM) for 4 h. All data are presented as the mean ± SEM from independent experiments performed in triplicate (*n* = 3). ### *p* < 0.001 vs. control; ** *p* < 0.01, and *** *p* < 0.001 vs. FA-treated group.

**Figure 2 pharmaceutics-15-01568-f002:**
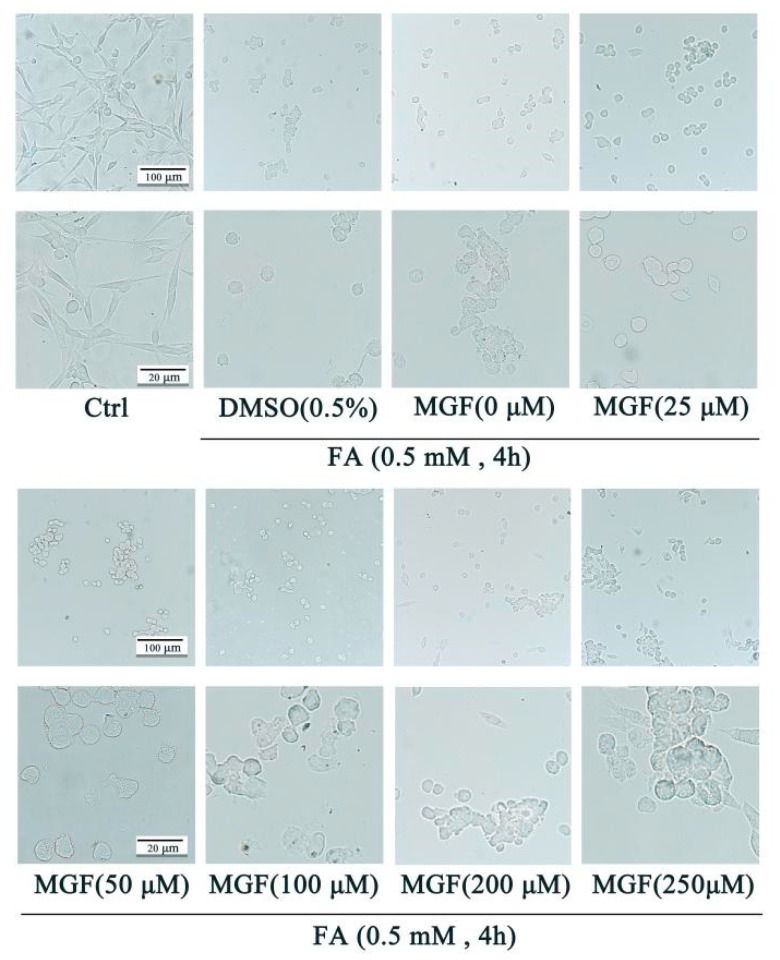
Protective effect of MGF against FA-induced cell morphological changes. HT22 cells were co-treated with MGF (25, 50, 100, 200, 200, and 250 µM) and FA (0.5 mM) for 4 h. Cell morphology was observed using a microscope at different magnifications (**upper** bar: 50 µM; **lower** bar: 25 µM).

**Figure 3 pharmaceutics-15-01568-f003:**
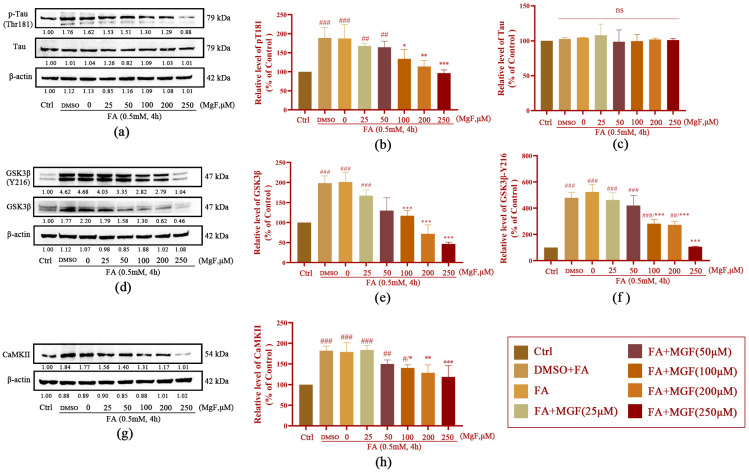
Effect of MGF on Tau, *p*-Tau (Thr181), GSK-3β, p-GSK-3β (Tyr216), and CaMKII protein in HT22 cells treated with FA. (**a**) Western blotting was used to detect the expression of Tau and p-Tau (Thr181) after FA (0.5 mM) and FA (0.5 mM) + MGF (25 to 250 µM) treatment; (**b**) p-Tau (Thr181) relative gray value analysis map; (**c**) t-Tau relative gray value analysis map; (**d**) Western blotting was used to detect the expression of GSK-3β and p-GSK-3β (Tyr216) after FA (0.5 mM) and FA (0.5 mM) + MGF (25 to 250 µM) treatment; (**e**) GSK-3β relative gray value analysis map; (**f**) p-GSK-3β (Tyr216) relative gray value analysis map; (**g**) Western blotting was used to detect the expression of CaMKII after FA (0.5 mM) and FA (0.5 mM) + MGF (25 to 250 µM) treatment; (**h**) CaMKII relative gray value analysis map; gray values (mean ± SEM) were calculated from independent experiments in triplicate (*n* = 3). # *p* < 0.05, ## *p* < 0.01, and ### *p* < 0.001 vs. control; * *p* < 0.05, ** *p* < 0.01, and *** *p* < 0.001 vs. FA treated group.

**Figure 4 pharmaceutics-15-01568-f004:**
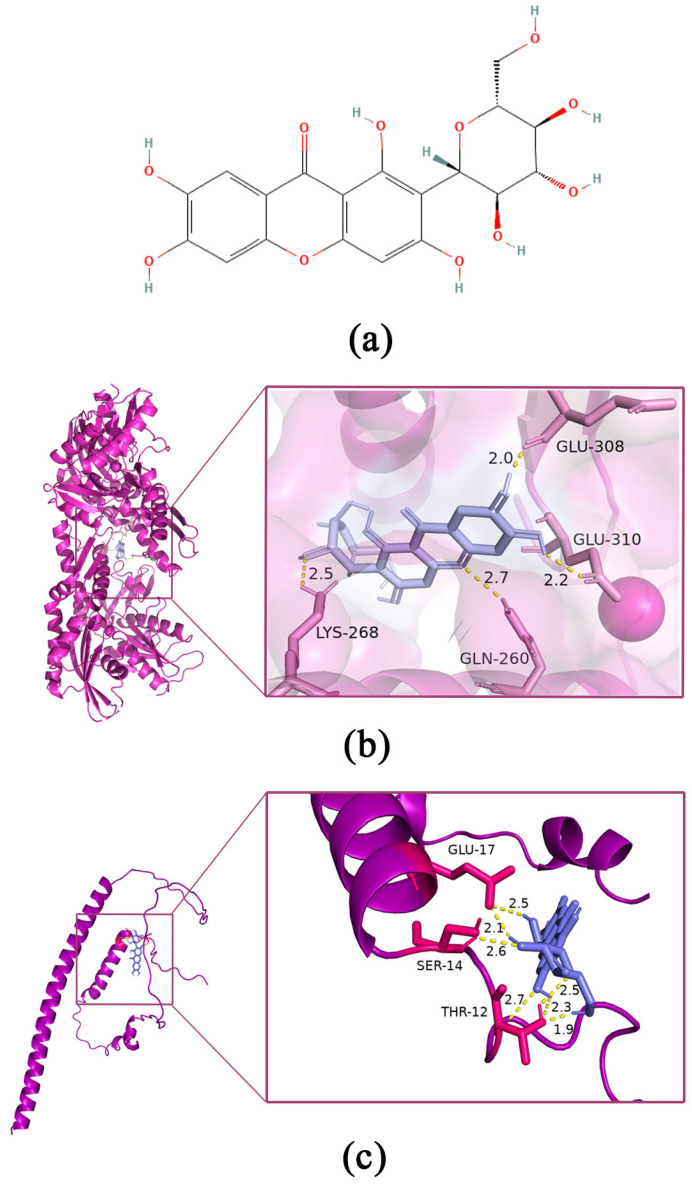
Molecular docking with ERS-related proteins using MGF as a small molecule ligand. (**a**) The 2D chemical structure of MGF (obtained from the PubChem database, CID: 5281647); (**b**) molecular docking between MGF and GRP78 with an intermolecular binding energy of −4.43 Kcal/mol and RMSD = 1.853 Å; (**c**) molecular docking between MGF and CHOP protein with an intermolecular binding energy of −2.79 Kcal/mol and RMSD = 0.173 Å (black numbers represent the hydrogen bond length at the docking site and the amino acid residue name).

**Figure 5 pharmaceutics-15-01568-f005:**
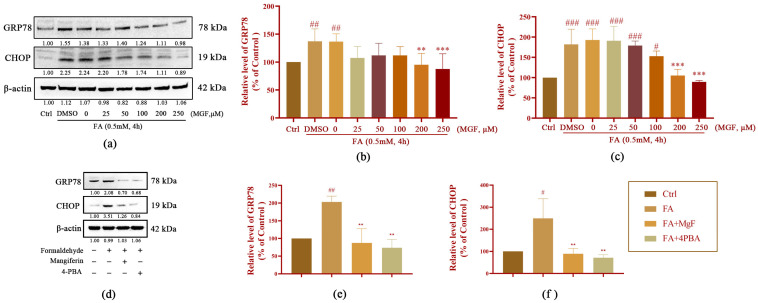
Effect of MGF on ERS-related proteins in HT22 cells treated with FA. (**a**) Western blot was used to detect the expression of GRP78 and CHOP after co-treatment with FA (0.5 mM), FA (0.5 mM), and MGF (25 to 250 µM); (**b**) Western blotting-based analysis of the relative gray values of GRP78; (**c**) Western blotting-based analysis of the relative gray values of CHOP; (**d**) Western blotting was used to detect the expression of GRP78 and CHOP after co-treatment with FA (0.5 mM), FA (0.5 mM), and MGF (25 to 250 µM) and FA (0.5 mM) + 4-PBA (1 mM); (**e**) Western blotting-based analysis of the relative gray values of GRP78; (**f**) Western blotting based-analysis of the relative gray values of CHOP; gray values (mean ± SEM) were calculated from independent experiments in triplicate (*n* = 3). # *p* < 0.05, ## *p* < 0.001, and ### *p* < 0.001 vs. control; ** *p* < 0.01, and *** *p* < 0.001 vs. FA-treated group.

**Figure 6 pharmaceutics-15-01568-f006:**
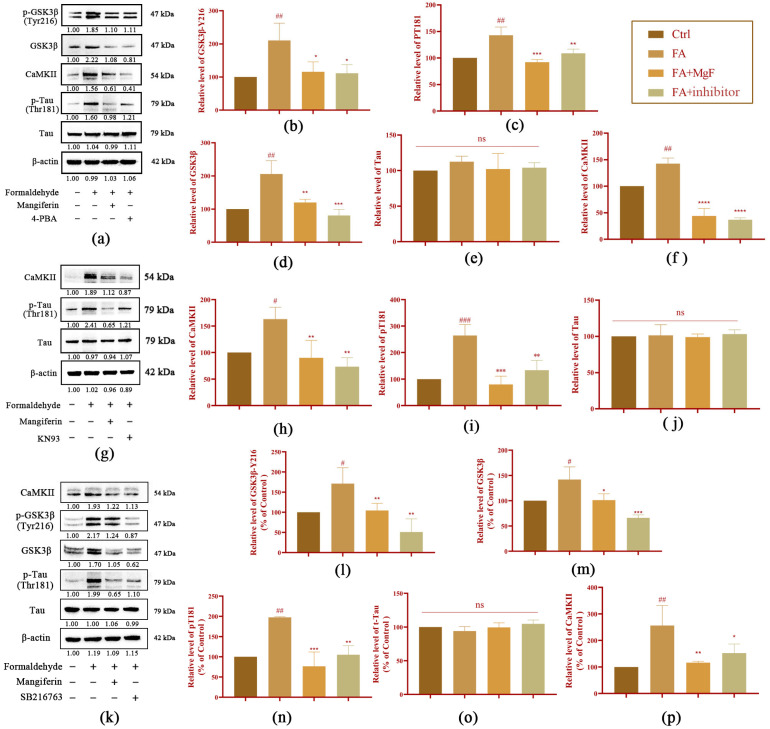
Effect of different inhibitors on Tau, p-Tau (Thr181), GSK-3β, p-GSK-3β (Tyr216), and CaMKII protein in HT22 cells treated with FA. (**a**) Western blotting was used to detect the expression of Tau, p-Tau (Thr181), GSK-3β, p-GSK-3β (Tyr216), and CaMKII protein after the FA (0.5 mM) and FA (0.5 mM) + 4-PBA (1 mM) treatment; (**b**) relative gray value analysis of p-GSK-3β (Tyr216); (**c**) relative gray value analysis of GSK-3β; (**d**) relative gray value analysis of CaMKII; (**e**) relative gray value analysis of p-Tau (Thr181); (**f**) relative gray value analysis of Tau; (**g**) Western blotting was used to detect the expression of Tau, p-Tau (Thr181), and CaMKII protein after FA (0.5 mM), FA (0.5 mM) + MGF (250 µM), and FA (0.5 mM) + KN93 (5 μM) treatments; (**h**) CaMKII relative gray value analysis map; (**i**) p-Tau (Thr181) relative gray value analysis map; (**j**) t-Tau relative gray value analysis map; (**k**) Western blotting was used to detect the expression of Tau, p-Tau (Thr181), GSK-3β, p-GSK-3β (Tyr216), and CaMKII protein after FA (0.5 mM), FA (0.5 mM) + MGF (250 µM), and FA (0.5 mM) + SB216763 (20 µM) treatments; (**l**) p-GSK-3β (Tyr216) relative gray value analysis map; (**m**) GSK-3β relative gray value analysis map; (**n**) p-Tau (Thr181) relative gray value analysis map; (**o**) t-Tau relative gray value analysis map; (**p**) CaMKII relative gray value analysis map; gray values (mean ± SEM) were calculated from independent experiments in triplicate (*n* = 3). # *p* < 0.05, ## *p* < 0.01, and ### *p* < 0.001 vs. control; * *p* < 0.05, ** *p* < 0.01, *** *p* < 0.001, **** *p* < 0.0001 vs. FA-treated group.

**Figure 7 pharmaceutics-15-01568-f007:**
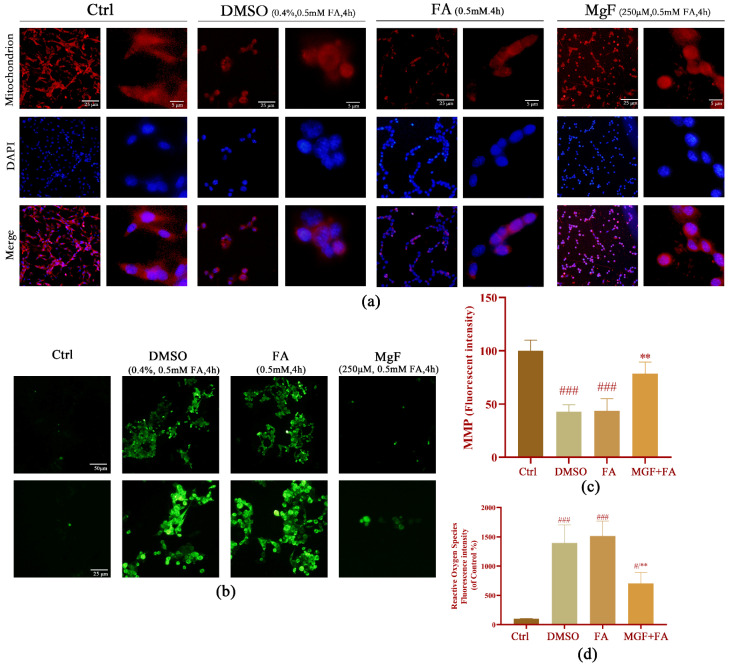
Effect of MGF on FA-induced oxidative stress in HT22 cells. (**a**) Mitochondrial fluorescence (red) changes in HT22 cells treated with DMSO (0.4%), FA (0.5 mM), and FA (0.5 mM) together with MGF (250 µM) for 4 h; scale bar, 25 μm and 5 μm; (**b**) ROS fluorescence (green) changes in HT22 cells treated with DMSO (0.4%), FA (0.5 mM), and FA (0.5 mM) together with MGF (250 µM) for 4 h; scale bar, 50 μm and 25 μm; (**c**) corresponding mitochondrial fluorescence intensity analysis; and (**d**) corresponding ROS fluorescence intensity analysis. Each assay was repeated three times with 10 cells/coverslip per replicate. # *p* < 0.05, and ### *p* < 0.001 vs. control; ** *p* < 0.01, vs. FA-treated group.

**Figure 8 pharmaceutics-15-01568-f008:**
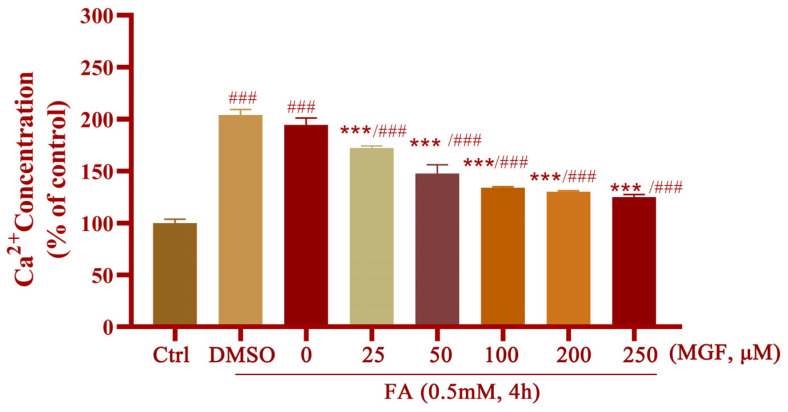
Protective effect of MGF on FA-induced Ca^2+^ dysregulation. All data are presented as the mean ± SEM from independent experiments performed in triplicate (*n* = 3)., ### *p* < 0.001 vs. control; *** *p* < 0.001 vs. FA-treated group.

**Figure 9 pharmaceutics-15-01568-f009:**
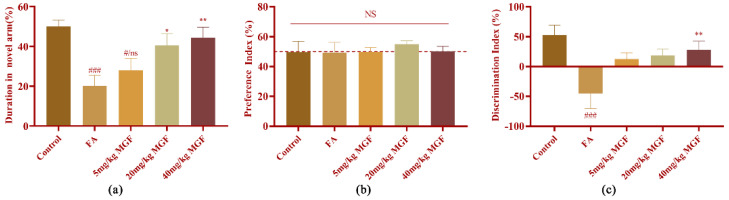
Effects of MGF on spatial memory and cognitive function in mice with FA-induced cognitive impairment using the Y-maze test and NOR test. (**a**) The total exploration time of the novel arm (duration in the novel arm) in the Y-maze was used to evaluate the spatial memory performance of the mice; (**b**) preference index of mice in the NOR test; (**c**) discrimination index was used to assess long-term memory in the NOR test. All data are shown as the mean ± SMD. *n* = 9–10/group. # *p* < 0.05, and ### *p* < 0.001 vs. control; * *p* < 0.05, and ** *p* < 0.01 vs. FA-treated group.

**Figure 10 pharmaceutics-15-01568-f010:**
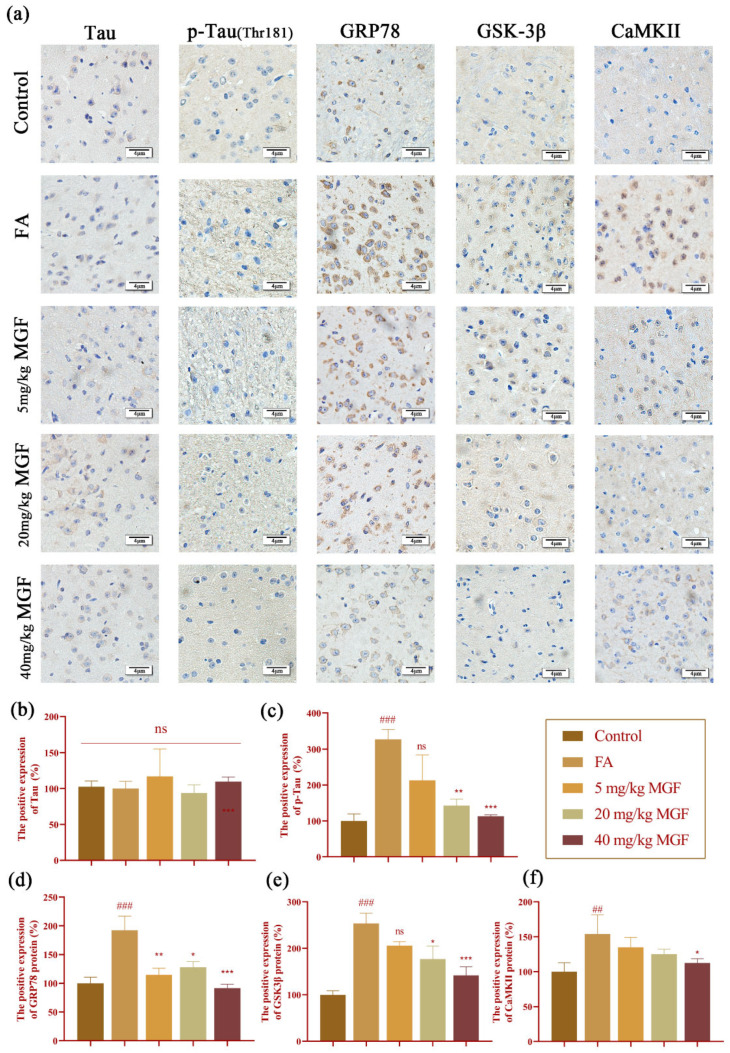
Expression of Tau, p-Tau, GRP78, GSK-3β, and CaMKII in the brains of mice. (**a**) Immunohistochemical staining for Tau, p-Tau, GRP78, GSK-3β, and CaMKII in the brains of mice. Scale bar, 4 μm. (**b**) positive rate of Tau expression; (**c**) positive rate of p-Tau expression; (**d**) positive rate of GRP78 expression; (**e**) positive rate of GSK-3β expression; (**f**) positive rate of CaMKII expression. ns: not significant; all data are shown as the mean ± SMD. *n* = 9–10/group. ## *p* < 0.01, and ### *p* < 0.001 vs. control; * *p* < 0.05, ** *p* < 0.01, and *** *p* < 0.001 vs. FA-treated group.

## Data Availability

Data are contained within the article.

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
