# Peer review of "The Protective Effect of Mangiferin on Formaldehyde-Induced HT22 Cell Damage and Cognitive Impairment"

_pharmaceutics, 2023, doi:10.3390/pharmaceutics15061568_

Round 1

Reviewer 1 Report

This study suggests that the polyphenol antioxidant MGF may have a neuroprotective effect against FA-induced neurotoxicity, using in vivo and in vitro approaches mice. MGF was found to improve spatial learning ability and long-term memory in mice, as well as protect against FA-induced cytotoxicity. These protective effects were achieved by attenuating FA-induced ERS and oxidative damage, including Ca2+ overload, ROS generation, and mitochondrial dysfunction. MGF was also found to inhibit the expression of downstream Tau-associated kinases GSK-3β and CaMKII. The authors suggests that MGF may have potential as a natural alternative for the treatment of AD. This study is interesting and shows a lot of potential, however, there are some issues that require a further look into the data, especially the lack of animal data for histological or molecular analyses.

Major:

1) Can you explain the rationale for choosing these specific concentrations of MGF used in the study for mice and cells?

2) The authors mention that MGF prevents tau hyperphosphorylation in HT22 cells, but what about in mice? How is tau phosphorylation in the brain of the MGF treated mice?

3) The authors mention that MGF did not significantly affect the expression of total Tau on cells, but how was this affected in the brains of mice?

4) It would be useful to discuss the potential limitations of using HT22 cells as a model system for studying neurotoxicity and the generalizability of the findings to other cell types.

5) The calcium overload experiments are not very convincing. Have the authors considered using additional assays to further support their findings on the protective effect of MGF against FA-induced oxidative stress and Ca2+ overload?

6) Why did the authors failed to conduct molecular and histological analyses in the brains of the mice that were used for the behavioural studies? What was the reason for this? At least the authors could have performed basic immunohistochemistry analyses. Were the results from mice contradictory to those obtained using the HT22 cells?

7) Considering that upon availability of animals, the authors chose to present results mostly with the HT22 cells: what are the potential limitations of using HT22 cells as a model for studying the protective effects of MGF in the context of neurological disorders? Is it a suitable model? This should be highlighted in your discussion.

8) The link between FA, ERS and oxidative damage in AD should be further explored in the text. Why is this relevant?

9) Additionally, the authors should further discuss the limitations of the study and propose future research directions for investigating the potential multi-system effects of MGF against FA toxicity.

Minor:

1) The protocol for Ca2+ measurement requires further detail, was a kit used? If so, there is no mention to its reference.

2) The manuscript could benefit from proofreading, as some statements are confusing and could be improved.

Reviewer 2 Report

Dear Dr.,

Title: The Protective Effect of Mangiferin on Formaldehyde-induced HT22 Cell Damage and Cognitive Impairment

Manuscript ID: pharmaceutics-2305903

Overall comments: Authors described in this manuscript: Mangiferin (MGF) effects against the formaldehyde (FA) induced neurotoxicity and cognitive impairments via reduction of oxidative damage, Ca2+ overload, ROS generations, and mitochondrial dysfunctions. However, the conclusion statement needs to revise in this manuscript. The overall manuscript is written well and it has novelty in this field of research.

Specific comments:

1.      In summary, the authors discussed the downstream process of Tau-associated kinases (GSK-3β and CaMKII) in HT22cell damage. The authors need to add information about mangiferin (MGF) effects against the formaldehyde (FA) induced neurotoxicity with reduction of oxidative damage, Ca2+ overload, ROS generations, and mitochondrial dysfunctions.

Minor comments

1.      In the abstract, the term ERS is not explained.

2.      Figure 7 can make clear with the expansion of the figure size is similar to other figures.

3.      References need to update with relevant recent references.

*****

Reviewer 3 Report

Dear Authors

The manuscript entitled "The Protective Effect of Mangiferin on Formaldehyde-induced HT22 Cell Damage and Cognitive Impairment" submitted by He et al., represents a novel approach to the study of Alzheimer's disease pathogenesis. To evaluate the influence of formaldehyde on the neurotoxic mechanisms of AD, highly innovative modern methods were used. In this introduction, all current concepts addressing the pathogenesis and molecular mechanisms affecting AD pathology are reviewed. The methods used in the study are described in sufficient detail. The results are presented clearly and correctly. The discussion is thorough and analyzes all current concepts concerning AD. 

I have a few minor remarks.

In the abstract it needs to be clarified what ERS means. Perhaps the acetylcholinergic theory of the development of dementia and the role of acetylcholinesterase inhibition in general in AD therapeutic regimens should be included in the introduction though. This, I think, is important to note briefly, especially since there are studies that have found that mangiferin inhibits AChE and improves cognitive function. I think it is good that the chemical structure of mangiferin is presented in the text.

Round 2

Reviewer 1 Report

The manuscript was greatly improved with the addition of data from mice. 

Nevertheless, the new text inserted requires proofreading. As an example, the final statement of the manuscript is somewhat confusing: 

"It should be 732 noted that, as an important environmental pollutant, toxicity caused by FA is not only 733 limited to neurological damage such as cognitive impairment and whether MGF can counteract FA-induced multi-system damage deserves further study in the future."
